# Perceived Barriers and Facilitators of Adventurous Play in Schools: A Qualitative Systematic Review

**DOI:** 10.3390/children8080681

**Published:** 2021-08-07

**Authors:** Rachel J. Nesbit, Charlotte L. Bagnall, Kate Harvey, Helen F. Dodd

**Affiliations:** 1College of Medicine and Health, University of Exeter, Exeter EX1 2LU, UK; h.dodd@exeter.ac.uk; 2College of Life and Environmental Sciences, University of Exeter, Exeter EX4 4QG, UK; c.l.bagnall@exeter.ac.uk; 3School of Psychology and Clinical Language Sciences, University of Reading, Reading RG6 6ES, UK; k.n.harvey@reading.ac.uk

**Keywords:** child, play, risky play, adventurous play, school, qualitative synthesis

## Abstract

Adventurous play, defined as exciting, thrilling play where children are able to take age-appropriate risks, has been associated with a wide range of positive outcomes. Despite this, it remains unclear what factors might aid or hinder schools in offering adventurous play opportunities. The purpose of this systematic review is to synthesise findings from qualitative studies on the perceived barriers and facilitators of adventurous play in schools. A total of nine studies were included in the final synthesis. The review used two synthesis strategies: a meta-aggregative synthesis and narrative synthesis. Findings were similar across the two syntheses, highlighting that key barriers and facilitators were: adults’ perceptions of children; adults’ attitudes and beliefs about adventurous play and concerns pertaining to health and safety, and concerns about legislation. Based on the findings of the review, recommendations for policy and practice are provided to support adventurous play in schools.

## 1. Introduction

Play is ubiquitous in childhood and is recognised by the United Nations Convention as a fundamental right of all children [1]. Importantly, as well as bringing feelings of happiness and joy, play has been associated with myriad benefits for children. In relation to physical health, play has been associated with increased physical activity [2,3,4], decreased sedentary behaviour [5,6], improvements in cardiovascular fitness [6,7], and reduced risk for childhood obesity [6,8]. Further, play supports the development of children’s physical abilities and is positively associated with their fundamental movement skills [9,10]. Play is also essential for children’s socioemotional health. Through play, children learn to share, negotiate, resolve conflicts, regulate their emotions, and control their impulses [11]. Cognitively, play supports children’s ability to make decisions and problem solve [12] and is associated with both learning behaviours and learning readiness [11,13,14].

Adventurous, or risky, play has been defined as exciting, thrilling play where the child experiences a level of fear and is able to take age-appropriate risks [15,16]. Sandseter [3] identified six categories of risky play; play at great heights, play at high speed, play with dangerous tools, play near dangerous elements, rough and tumble play, and play where children can disappear/get lost. Children appear to enjoy playing in this way [17] and feel strongly about being afforded opportunities to assess risk for themselves [18]. Despite this, there is evidence that children’s opportunities for, and engagement in, adventurous play has declined in recent decades. Children play outside less than in previous generations [19], have less independent mobility [20] and are not allowed out alone until they are almost two years older than their parents were [21]. These declines have often been attributed to increased societal concerns surrounding children’s safety [22]. Dodd and Lester’s [15] conceptual model of adventurous play and anxiety highlights the critical role of the social environment in facilitating adventurous play. Specifically, the authors argue that the nature of adult supervision and rules and policies constrain children’s opportunities to take risks in their play [15]. These factors are likely to play a role in children’s adventurous play both in and out of the school context. Adults, as such, represent an important constraint on children’s opportunities to take risks and challenges in their play [23,24]. Indeed, it is known that parents’ attitudes and beliefs about risk during play are associated with the amount of time children spend playing in an adventurous way [21].

There are concerns for what this decline in adventurous play may mean for children’s health broadly, and in particular for their mental health. Specifically, Gray [11] argues that that the decline of children’s play may be a contributing factor to the rise of mental health problems in children and adolescents [25]. Alongside this, there are concerns that a culture of risk aversion may limit children’s risk taking and, in doing so, deny them the opportunity to learn from these experiences, affecting their ability to effectively judge risk in adolescence and into adulthood [26]. Indeed, it has been proposed that children’s engagement in age-appropriate risk through adventurous play may provide an adaptive means by which children can learn about fear, uncertainty, risk judgement, and coping. This learning may act as a protective factor for children in later life when they are faced with situations that provoke fear or uncertainty [15].

Whilst theoretical work on adventurous play is in its infancy, there is a growing body of empirical research demonstrating that adventurous play may be beneficial for other aspects of children’s health. For example, a systematic review conducted in 2015 examined the relationship between risky outdoor play and a range of health outcomes [27]. The authors concluded that environments that supported risky play had a range of benefits for children’s health, behaviour and development, including increased physical activity, decreased sedentary behaviour, improvements in social interactions and reported improvements in creativity and resilience in children. Although there are understandable concerns about child injuries, unstructured play, defined as play that is spontaneous, self-directed, intrinsically motivated and with an absence of external rewards [11,28], is relatively low risk (0.15–0.17) when compared to the incidence rates of injury per 1000 h for sports (0.20–0.67) and active transportation (0.15–0.52) [29]. The outcomes of the 2015 review informed the publication of an international position statement on outdoor active play in children aged 3–12 years [30]. This states that “Access to active play in nature and outdoors—with its risks—is essential for healthy child development.” [30] (p. 1).

A number of school-based interventions focussed on increasing children’s opportunities for adventurous, or risky, play during recess or breaktimes have been designed and some have been evaluated. In several instances, these have consisted of introducing loose parts into the play space [31,32]. Loose parts are materials with no fixed purpose (e.g., a tyre, boxes) and have been found to afford children the opportunity to take risks in their play [33]. The Sydney Playground Project focussed on the introduction of recycled materials into school playgrounds and included a risk-reframing workshop for parents and teachers [31]. In the UK, Outdoor Play and Learning (OPAL) provide support to help schools improve play during breaktimes, which includes addressing barriers around risk aversion [34]. Further, in New Zealand, the PLAY study focussed on increasing opportunities for risk and challenge, reducing rules, and adding loose parts [32]. Where these programmes have been evaluated, findings show that increasing children’s opportunities for adventurous play increases children’s physical activity [35]; although not consistently across studies [36], decreases disruptive behaviour and benefits children’s learning and social development [34], increases creativity and resilience [35], and improves children’s happiness at school [32]. Thus, adventurous play may be beneficial for various facets of children’s health.

Although there is initial evidence for positive effects of play-based interventions, with a risky play component, the extent to which they build upon evidence regarding what needs to be targeted to increase opportunities for, and engagement in adventurous play in schools is unclear. It is therefore essential to understand the barriers and facilitators that schools face in providing opportunities for (i.e., to afford the environment for adventurous play) and allowing engagement in (i.e., to provide permission to engage in) adventurous play, in order to design optimal and effective interventions. Our aim in this review is therefore to bring together findings from qualitative research providing insights into the perceived barriers and facilitators of adventurous play in schools. To analyse the findings yielded via a systematic search, two review methodologies were used. Papers that met a pre-specified quality threshold were analysed via meta-aggregative synthesis and papers that met our inclusion criteria but did not meet the quality criteria were analysed via narrative synthesis. The results of each will be presented separately and brought together in the discussion.

## 2. Methods

### 2.1. Inclusion Criteria

Prior to conducting the search, the search strategy protocol was published on the Open Science Framework (it is available to view here https://osf.io/34hfp/). To briefly summarise, the work had to meet the following criteria to be considered for inclusion. The work had to concern play for school-aged children, excluding work conducted in early years settings (pre-school provisions) and forest schools. School starting ages differ by country and we wanted to focus on the formal school context given that play is often already embedded within pre-school early years curriculums [37]. Forest schools were excluded because they typically run either independent to regular schools or are offered for a defined period of time rather than embedded in the school week and part of children’s day to day experience. Studies were required to be about play that took place during normal school hours, excluding after school clubs or residential trips. The work was required to make reference to adventurous play or risky play (for full terms https://osf.io/34hfp/). Further, the research was required to reference attitudes, perceptions, feelings, beliefs, experiences, barriers, or facilitators towards adventurous/risky play within the school context. For inclusion, the research had to be a qualitative or a mixed methods study where the qualitative work could be isolated from any quantitative data analysis. Reviews and theoretical papers were excluded. Articles were not required to be published in a peer-reviewed journal. Given that a substantial proportion of play literature is not peer reviewed, we felt that a search of the grey literature was critical to ensure that we included important, non-academic work that could inform our understanding of the perceived barriers and facilitators for children’s adventurous play in schools.

### 2.2. Search Strategy and Study Selection

Databases that were searched included PubMed, PsychINFO, Web of Science, ERIC, EThOS, ProQuest and Google Scholar. In addition, we undertook hand-searching of relevant stakeholder organisations’ websites and requested relevant work from contacts directly. The following search terms were used. Search terms relating to the setting: (school) AND search terms relating to adventurous play: (“adventurous play” OR “risky play” OR “challenging play” OR “risk taking in play” OR “play with risk” OR “risk in play” OR “rough and tumble play”) AND search terms relating to the evaluation (of adventurous play): (attitudes OR “perception*” OR barriers OR facilitators OR feelings OR belief* OR experience) AND search terms relating to the study design: (qualitative OR interview* OR “focus group*” OR ethnograp*).

The search was managed via Covidence software [38]. A total of 1712 articles were identified via the main search with a further 23 articles identified through other sources (e.g., requested work from contacts, hand-searching of relevant organisations’ websites), giving 1735 studies that were imported into Covidence [38]. Of these, 31 duplicates were removed (see Figure 1). All title and abstracts were screened by two independent reviewers (R.J.N. and C.L.B.). Studies that did not meet the inclusion criteria were discarded. Agreement between reviewers was good (97% agreement). Any disagreements between the reviewers were discussed. Following in-depth discussions, if either reviewer selected the study for potential inclusion, the article was included for full-text screening. Forty-nine articles were assessed against the inclusion and exclusion criteria at the full-text level, again each by two independent reviewers (R.J.N. and C.L.B.). Agreement between reviewers was good (96%). Two articles were discussed with a third reviewer (H.F.D.) to reach consensus. Forty-one of these articles were excluded for not meeting the criteria (see Figure 1). The remaining 8 papers were screened and forward and backward citation searching was conducted. From this, one further article was identified as being relevant by both reviewers and was included in the final selection. Nine articles were identified as meeting the criteria for eligibility for inclusion within the review.

### 2.3. Quality Assessment

The nine articles were assessed for quality using the JBI Critical Appraisal toolkit (see https://joannabriggs.org/sites/default/files/2019-05/JBI_Critical_Appraisal-Checklist_for_Qualitative_Research2017_0.pdf (accessed on 1 December 2020) by two independent reviewers (R.J.N. and C.L.B.). Agreement between reviewers was good (86%). Any disagreements were resolved through further in-depth discussions and re-reading of the texts in full; where consensus was not reached the specific criteria was recorded as unclear. The outcome of the quality appraisal determined whether each article was analysed using meta-aggregative synthesis or narrative synthesis. The quality criteria for the meta-aggregative synthesis were decided in advance. Any articles not meeting these criteria were considered for inclusion in the narrative synthesis. To be included in the meta-aggregative synthesis, studies must have used a qualitative approach to design, data collection, and analysis and demonstrate congruity between the research methodology and the representation and analysis of data (criterion 4); acknowledge the influence of the researcher on the research and vice versa (criterion 7); demonstrate that participants’ voices were adequately represented (criterion 8); and evidence that the conclusions drawn in the research report flowed from the analysis or interpretation of the data (criterion 10). To be included in the narrative synthesis, articles needed to show evidence that the conclusions of the research flowed from analyses and interpretation of the data (criterion 10) only.

Following the critical appraisal of the nine articles, four articles met the criteria for the meta-aggregative synthesis, with the remaining five articles meeting the criteria for the narrative synthesis (see Table 1 for critical appraisal results).

### 2.4. Meta-Aggregative Synthesis

#### 2.4.1. Date Extraction

A meta-aggregative approach developed by the Joanna Briggs Institute [46] was followed to synthesise the findings. Data extraction in a meta-aggregative review is a multi-phase process. The first phase consists of extraction of the general details of the studies, including information on the setting, cultural, participant characteristics, phenomenon of interest, methods, and analytical approach. The second phase consists of the extraction of the findings.

During the extraction phase, only findings relevant to the review question were extracted. A finding refers to an extract of the authors’ interpretation of their results. The findings here were not limited to the themes of the papers but were extracted from the repeated reading of the text; this decision was made to account for overarching themes that consisted of many sub-themes relevant to the review question. Each extracted finding was accompanied by an illustration where possible: a participant voice, fieldwork observations, or other available data. Findings and illustrations were extracted by two reviewers (R.J.N. and C.L.B.). Each finding was allocated a level of credibility, to indicate the extent to which the finding was supported by the available data. Each finding was judged as either unequivocal (findings that are supported by the data beyond reasonable doubt), credible (findings with unclear association with the data and open to challenge), or unsupported (findings that are not supported by available data). Findings that were not supported were not included with the data synthesis.

#### 2.4.2. Date Synthesis

Following the extraction of findings from the four studies, categorisation began. Categories were developed by bringing together two of more findings that were similar in meaning. Categorisation of findings was led by one reviewer (R.J.N.) who discussed the categorisation with the wider research team, which led to further refinement of the categories. The categories were next aggregated to form synthesised findings, containing at least two categories and which provided an overarching description of a group of categories.

### 2.5. Narrative Synthesis

The remaining five articles, that did not meet the quality criteria for inclusion within the meta-aggregative synthesis, were analysed using a narrative approach. This process consisted of reading the results of the studies in full, extracting findings and themes, and drawing together findings between and across the studies. In some instances, where the findings referred to higher-order themes, repeated reading of the text allowed the researchers to extract subthemes.

## 3. Results

### 3.1. Meta-Aggregative Synthesis

#### 3.1.1. Description of Included Studies

Four articles met the criteria for inclusion within the meta-aggregative synthesis (see Table 2 for details). Two of these studies were conducted in Australia [42,43], one in Turkey [39], and one in the UK [45]. Three of the four articles were published in peer-reviewed journals [25,26,27]; the remaining article was a Masters thesis [45].

Three of the four studies described qualitative work in relation to interventions. One study focussed on the pre-service teachers (i.e., student teachers in training) who were taking part in a six-week intervention designed to change their understanding of children’s risky play [39]. Two of the articles [42,43] were related to The Sydney Playground Project (SPP)—a cluster randomised control trial designed to evaluate the effectiveness of a programme that aimed to change adult views around managing risk taking in play. Spencer et al. [42] examined teachers’ sense-making in the management of risks with children with disabilities and Sterman et al. [43] examined educators’ experience of participating in the SPP more broadly. Wright [45] focussed on examining UK headteachers’ attitudes towards and perceptions of risky play.

A range of methods were used within the studies (see Table 2). Cevher-Kalburan [39] used a questionnaire consisting of open-ended questions to assess views of risk, risk taking, risks children may take during play, the benefits and hazards of risky play, and the role of adults and the physical environment in children’s play prior to the intervention. Data were also collected from researcher’s reflective notes after each session of the intervention course, participants’ brief evaluations at the end of the intervention, and participant drawings of their ideal playgrounds.

Spencer et al. [42] used qualitative responses elicited from the Tolerance of Risk in Play Scale (TRiPS) consisting of closed and open-ended questions about teachers’ perspectives on risk and risk taking by children with disabilities. Researchers’ field notes and video recordings of children’s play were also collected.

Sterman et al. [43] and Wright [45] used semi-structured interviews to examine school staff’s experiences of participating in the SPP, and headteachers perceptions of risky play, respectively. In addition, Wright [45] employed photo-elicitation during the interviews; a technique used where photos or other visual material are used to stimulate verbal discussion [47].

The participants of three of the studies were school staff including teachers, teaching assistants, therapists, and school leadership. The participants in Cevher-Kalburan [39] were all pre-service teachers.

Three of the four studies used a thematic approach to analysis [42,43,45]. Cevher-Kalburan [39] used content analysis coupled with participants’ qualitative brief evaluations.

#### 3.1.2. Data Synthesis

We extracted 99 findings from the four articles. Of these, 69 were judged as unequivocal or credible and were included in the categorisation phase. The 69 findings were brought together into 13 categories. The 13 categories were aggregated into five synthesised findings (see Figure 2), which are detailed below.

### 3.2. Synthesised Finding 1: External Judgements and Legislative Factors

This finding included three categories, detailed below. See Table 3 for illustrations in support of each category.

#### 3.2.1. Accountability Guides Supervision on the Playground

This category relates to how the behaviour of school staff on the playground was limited by accountability to children, parents, other educators and themselves. Within the studies, staff reported that “keeping students safe” guided all aspects of supervision at school [43] and that unsupervised play was not possible because of concerns pertaining to duty of care [42].

#### 3.2.2. Fear of External Judgement

This category captures staff reports of fearing external judgement as a consequence of offering adventurous play opportunities. These external judgements were made in reference to outside agencies (e.g., Ofsted [Ofsted is the Office for Standards in Education, Children’s Services and Skills, a regulatory body and department of the UK government responsible for inspecting educational institutions that care for children and young people]) perceiving adventurous play as a “waste of time” [45] and from passers-by judging the aesthetics of the school playground [43]. 

#### 3.2.3. Perceived Consequences of Failing in Duty of Care

The main focus of this category was concerns regarding failing in duty of care, often in relation to injuries resulting from adventurous play activities. Within this category, participants cited fear of censure or condemnation in the event of an accident [45] as well as concerns about the consequences of an injury such as loss of professional accreditation [42] and challenges communicating with other staff [45] and parents [42,43].

### 3.3. Synthesised Finding 2: Perceptions of Children

This finding included two categories, detailed below. See Table 4 for illustrations in support of each category.

#### 3.3.1. Children as Unable to Judge Risk and Initiate Play

This category describes how staff perceive children; in particular, there is reference to perceptions of children as unable to judge risk or initiate play and children’s lack of understanding about the potential harms in their play. Notably, this category encompassed findings from two articles, both of which focussed on children with disabilities. This category therefore may not reflect adult’s perceptions of children without disabilities.

#### 3.3.2. Recognition of Children’s Play and Their Ability to Play

This category focusses on staff recognition of children’s ability to play and use their personal agency [42], be interested in adventurous play activities [43], and assess risk for themselves [39].

### 3.4. Synthesised Finding 3: Stepping in and Stepping Back

This finding included two categories, detailed below. See Table 5 for illustrations in support of each category.

#### 3.4.1. Staff Intervening and Directing Children’s Play

The main focus of this category was evidence and discussion of specific actions that staff engage in to manage the uncertain nature of children’s play. Within this category there were examples of staff intervening and directing children’s play in order to prevent potential harms before they emerged [42]. Evidence revealed that uncertainty in play is often shut-down [42,45], either by removing children from the play situation [42], or adults’ direction through speech; e.g., “You can’t do that”, “I know you’re having fun but you need to keep your body safe” [45].

#### 3.4.2. Staff Stepping Back

This category encompassed findings relating to how staff “stepping back” facilitated children’s engagement in adventurous play. In Sterman et al. [43], staff were explicitly instructed to “step back” which included not warning children about dangers or directing their play and observing what children would do without staff input. “Stepping back” in this context therefore refers to specific actions staff took to not intervene or direct children’s play. This is often central to interventions for supporting children’s play because it gives children space to explore and evaluate risk for themselves; often adults intervene too early and remove the opportunity for children to do this independently. It was evident from the studies that stepping back allowed staff to recognise children’s abilities and that through recognising children’s abilities this enabled staff to take a step back and let children play [43]. It was also clear that school staff were aware that they needed to make a conscious decision to step back and not intervene in children’s play [42].

### 3.5. Synthesised Finding 4: School Environment and Culture

This finding included three categories, detailed below. See Table 6 for illustrations in support of each category.

#### 3.5.1. Supporting Adventurous Play through Training

This category was specific to Wright, [45] and covers findings relating to the importance of training and risk assessment processes to enable risky play activities [45], as well as the importance of empowering school staff to facilitate risky play. The category includes findings endorsing the support of external agencies (Local Authority Children Health and Safety and Wellbeing Manager), and the need for lunchtime supervisors to understand the value of risky play.

#### 3.5.2. Importance of Parent Support

This category includes findings relating to the importance of parent support for adventurous play. In particular, it was evident that there was a need for parental collaboration and support [45]. In Sterman et al. [43], staff cited that having parents present in risk-reframing sessions gave educators more freedom to allow children independence on the playground, and that this mutual understanding acted to mitigate some of the fears relating to accountability to parents. Indeed, when parents were not present in the risk-reframing sessions, schools voiced this as a “key miss”, stating that it would have been better if parents were present [43]. These findings suggest that parent support and collaboration is valued by staff [43].

#### 3.5.3. Practical Considerations

Although not widely discussed, this category included two findings pertaining to the practicalities of adventurous play, namely time constraints for implementing adventurous play and the importance of constant reviewing and assessment of the play conditions, for example assessing the impact of the weather on the assessment of risk of the play materials [45].

### 3.6. Synthesised Finding 5: Perceptions of Adventurous Play

This finding included three categories, detailed below. See Table 7 for illustrations in support of each category.

#### 3.6.1. Positive Beliefs and Commitment to Adventurous Play

This category captured positive beliefs and commitment to adventurous play. Within this category positive beliefs were often held in reference to understanding of the benefits of adventurous play [39,45] and the need for commitment for adventurous play in schools [45].

#### 3.6.2. Uncertainty and Anxiety Surrounding Adventurous Play

This category describes findings referencing the uncertainty of supervising adventurous play. It was expressed that lunchtime play supervisors were thought to limit adventurous play opportunities owing to their misperception of adventurous play as behavioural issues [45]. For example, one headteacher suggested that lunchtime supervisors are not quite sure if the play is moving to an unacceptable level of behaviour (see Box 5). Within this category was also reference to uncertainty of the role of staff in supervising adventurous play and uncertainty about knowing how to act on the “step back” message they were given during the intervention [43]. This category also encompassed the anxiety of staff in supervising adventurous play, such as adults that would prefer not to grant risky play opportunities due to their natural tendencies to be risk averse [45].

#### 3.6.3. Perceptions of Risk

This category focussed on perceptions of risk, both generally [45] and changes in perceptions of risk following interventions [39]. The category also encompasses perceptions of play equipment, for example fixed play equipment was perceived as less risky as it was an “accepted object”.

### 3.7. Narrative Synthesis

#### 3.7.1. Description of Included Studies

Five studies were included in the narrative synthesis (see Table 8), four of which were peer-reviewed journal articles and one was a non-academic published evaluation report [34]. The research was conducted in New Zealand [32], Sweden [40], Australia [41], the Netherlands [44], and the UK [34].

Three of the five reports described qualitative work in relation to interventions. One focussed on experiences of school leaders participating in a randomised control trial that implemented an intervention designed to increase risk and challenge in the school playground [32]. One focussed on evaluating a programme which aimed to enable schools to offer challenging and exciting play opportunities for children [34]. Finally, Niehues et al. [41] focussed on a risk-reframing intervention offered to parents and educators to change perceptions of risk in children’s outdoor free play. The remaining two studies examined teachers’ perceptions of risk and safety in children’s risky outdoor play [40] and professional attitudes towards children’s risky play [44].

The participants in the studies were primarily educators and professionals working in schools [41,44], which included school leaders [32] and teachers [40]. In Niehues et al. [41], parents also participated in risk-reframing sessions alongside educators. The methods of data collection varied, and often multiple methods were used within the same study (see Table 3). The methods included qualitative interviews [32,34,41], questionnaires with open-ended responses [44], focus groups [34,40], observations [34,40], brief evaluations [41], and recording of risk-framing sessions [41]. To analyse data, two of the studies used thematic analysis [32,34], one study used a social analysis [41], and two studies used content analysis [40,44].

The primary reason these five studies were excluded from the meta-aggregative review related to the JBI Critical Appraisal Toolkit criterion (see https://joannabriggs.org/sites/default/files/2019-05/JBI_Critical_Appraisal-Checklist_for_Qualitative_Research2017_0.pdf (accessed on 1 December 2020), requiring acknowledgement of the influence of the researchers on the research and vice versa (criterion 7) [32,34,40,41,44]. For two of the studies, the congruity between the research methodology and the representation of the data were rated as unclear (criterion 4) [34,44] and for one of the studies there was lack of evidence to demonstrate that participants’ voices were adequately represented (criterion 8) [44] (see Table 1 for critical appraisal ratings).

A narrative synthesis relies primarily on the use of words to explain the findings of a group of papers. Findings were extracted by two independent reviewers (R.J.N. and C.L.B.). Following the extraction of relevant findings, the data were translated, drawing together the primary themes or concepts reported across the studies. Through repeated reading of the text, the themes and sub-themes were aggregated into overarching categories of findings with similar meanings. This inductive approach was used to organise the findings and summarise the main themes across the studies. Our analysis of the findings of these studies led to the development of nine themes, which are detailed below.

#### 3.7.2. School Dynamics

Several of the studies suggest that the dynamics within the school institution can act as barriers for opportunities for adventurous play in schools [32,34,44]. In particular, this was often with reference to the differing opinions and characteristics of school staff that made it difficult to reach agreement with regard to adventurous play opportunities [44]. In Farmer et al. [32], a headteacher stated that it was not as easy as just making a decision and referred to the school as a “democracy” and stated that “this becomes very hard when everybody, … think they’ve got an input”. Similarly, in Lester et al. [34], the headteacher references school staff with different values, experiences, and approaches and stated that the hardest challenge is to “get the whole school community to do it with you” (including staff, parents, and children). Lester et al. [34] highlighted the importance of developing a whole-school approach to adventurous play. Similarly, in Van Rooijen et al.’s [44] study, there was mention of the need to gain the moral support of all school staff.

The need to get the school community involved in change was not exclusive to lunchtime staff and teachers, but also encompassed the wider school community. In Farmer et al. [32], it was mentioned that it could be difficult to get caretakers and teachers to change their attitudes towards play. Within this study, one school leader made reference to the decision to stop mowing wilderness areas, which took “quite a bit of persuading, because the guy on the tractor just couldn’t abide seeing the area not mowed”.

Studies described that shifting attitudes to children’s play requires strong leadership, [34] and that whilst a team effort and shared goal was needed to implement change, it was also necessary to delegate tasks [32]. Relatedly, there was concern about whether the plan (of change) could be realised if a team member considered the “driving force” left.

#### 3.7.3. Parent Support

Studies referenced the need for parents to support adventurous play and described perceptions of parents’ concerns as a barrier to schools providing adventurous play opportunities [32,41,44]. In Van Rooijen et al. [44], educational professionals discussed that parents being over-protective and anxious in relation to injuries and dirty or damaged clothing as a barrier of adventurous play in schools. Similarly, in Niehues et al. [41], teachers raised concerns that parents can be anxious about what might happen to their children. These findings highlight that staff appear to be concerned about parents’ perceptions of adventurous play.

In Farmer et al. [32], relaxing the rules and giving permission for adventurous play opportunities was said to create backlash from some parents, who as a consequence, moved their children to a different school. On the other hand, this same study showed that school leaders believed changes in play within the school encouraged children to come to the school. These findings highlight individual differences in parents’ acceptability of adventurous play provisions in schools.

#### 3.7.4. Perceptions of Children

Perceptions of children’s capabilities were salient in the studies [32,34,44]. On the one hand, there was evidence that staff perceived children as unable to see risks or overestimate their abilities when engaging in risky play [44]. For example, in Niehues et al. [41], a teacher explained how children are not allowed to take any risks because they are a “precious cargo”, demonstrating how perceptions of children may limit risk-taking opportunities granted in the playground.

On the other hand, Lester et al. [34] described that there was a gap between adults’ expectations of how children would use the space and materials in their play and how children actually used the space and materials. Lester et al. [34] reported that staff gained an understanding of children’s willingness to experiment in their play and play in ways that staff did not realise were possible. This recognition of children’s capabilities was also evident in Farmer et al. [32] where, following participation in an intervention designed to increase risk and challenge in the playground, teachers recognised that children could “handle themselves” in their play and were surprised at children’s confidence and skills in their play.

#### 3.7.5. Attitudes and Beliefs about Adventurous Play

In several of the studies, positive beliefs towards risk in play were evident [40,44]. These positive attitudes included the belief that risks are everywhere and cannot be avoided [40], that accidents rarely happen [40], and that exposure to risk is beneficial to children’s development [40,44]. There was also the belief that adults did not want to cultivate caution in children but instead wanted them to test their limits even if this meant the child could get injured. These findings were coupled with the perception that the outdoor play environment should not be too protective as it might inhibit children’s development [40].

It was clear that even where attitudes towards children’s adventurous play were not positive, they could be changed during interventions [32,34]. Interventions were reported to have challenged perceptions of adventurous play and lead to a “huge shift” in perceptions of health and safety, as well as a “big ethos change” towards accepting that the benefits of adventurous play cannot be realised without some risk taking [34]. Similarly, in Farmer et al. [32], staff appeared to reflect on whether there is “really a good reason for saying no” and there was evidence of “letting go” of things that had once been held as important, which created a more permissible play environment. For example, this included relaxing of the rules and accepting that the school playground at times could look messy. Positive attitudes towards adventurous play therefore appear fundamental in allowing children the opportunity to engage in adventurous play.

The necessity to establish a balance between letting children take risks coupled with the requirements for careful supervision was mentioned [44]. In Van Rooijen et al. [44], staff experienced uncertainty about balancing their own positive attitudes towards children’s adventurous play and how to act on this in practice against a backdrop of regulations, protocols, and policies. There was also evidence that staff’s own experiences influenced how they set limits on children’s play [40]. It was evident that staff recognised that although they had the best intentions, they often became barriers to children’s risk-taking opportunities [41].

These barriers often reflected staff perceptions of the uncertainty of adventurous play, including perceptions of risk and uncertainty pertaining to supervising adventurous play. In Gyllencreutz et al. [40], fixed play equipment was perceived as safer, even if the height and surface beneath were the same, implying differences in perceptions of risk for equipment that affords children to play in an adventurous way. In Van Rooijen et al. [44], personal barriers were reported to impact individual staff member’s approaches to the supervision of adventurous play and were related to feelings of tension, fear, and doubt on when to intervene in play. Supervisors reported finding it difficult to guard the boundaries for children and were concerned about the possibility of giving approval of unacceptable risk in the playground [44]. This theme was also present in Niehues et al. [41]; educators were shown to share feelings of uncertainty, demonstrating concerns that children were pushing limits of acceptable risk.

Despite feelings of uncertainty and anxiety in supervising children’s play, Niehues et al. [41] gave an example of an educator wanting to intervene in the play, but intentionally tolerating the uncertainty in order to support children’s risk-taking. This notion of tolerating uncertainty was also evident in Lester et al. [34], where the headteacher described her anxiety in allowing adventurous play to take place and the need to tolerate the uncertainty of the play.

#### 3.7.6. Accountability, Duty of Care, and the Perceived Negative Consequences of Adventurous Play

In Lester et al. [34], lunchtime supervisors discussed that despite recognising the key principles of training that were provided, there were still concerns about accountability. This was also evident in Niehues et al. [41], where teachers cited duty of care, the worry of the consequences if parents disagreed with their decisions, and the risk that they might lose their jobs if parents complained. These ideas were further present in Van Rooijen et al. [44] where the “undesirable” effects on other children in their play was mentioned and how the undesirable effects may trigger loss of “clientele” to the school. This was also coupled with the possibility of accidents, injuries, and serious harm to children [44].

Schools’ awareness of their safety responsibilities was cited as the reason by which an activity or cause of an injury or incident would be removed [40], emphasising how concerns about accountability, duty of care, and the perceived negative consequences of adventurous play may impact the activities provided.

#### 3.7.7. Regulations and Policies

Barriers also pertained to safety regulations, although this was only described in one of the papers. In Van Rooijen et al. [44], staff felt dissonance between their positive attitudes towards risky play and the restrictions they experience. This study also mentioned that staff believed regulations, protocols, and policies needed to be less strict and more generous in offering opportunities for risky play. This idea was coupled with the viewpoint that consent needed to be given by health authority organisations for risky play activities to take place [44].

#### 3.7.8. Education and Training on Adventurous Play

Several studies mentioned the need for professionals to gain insight into, and experience with, children’s risky play [44]. In Farmer et al. [32], it was stated that, without education and training, schools typically had little knowledge about how their rules and practices were impacting on children’s play experiences. In other studies [34], it was stated that conversations and training from a play advisor helped change a member of staff’s thinking regarding adventurous play. It was apparent that conversations with play advisors and Health and Safety officers helped in reassuring the staff and providing peace of mind regarding litigation. The relationship with the play advisor was cited as pivotal in inspiring the staff, making changes, and “getting it off the ground”. In Van Rooijen et al. [44], study participants expressed that parents need greater insight into the value of risky play to reach agreement with educational professionals about opportunities for adventurous play at school.

#### 3.7.9. Practical Considerations

Several of the studies referred to barriers pertaining to the practicality of adventurous play in schools. Specifically, in response to factors important in supporting professionals in their approach towards children’s risk taking in play, Van Rooijen et al. [44] found that professionals believed that the outdoor play environments require additional risky play opportunities. There were also concerns about equipment not lasting, the storage and maintenance of equipment, and the time required to source loose parts to facilitate adventurous play [32]. Relatedly, financial barriers were described; in Van Rooijen et al. [44], some educational professionals mentioned the need for financial support for risky play facilities, and in Farmer et al. [32], school leaders expressed that there was not enough money available to make all the changes they wanted to the play environment.

#### 3.7.10. Giving Permission for Adventurous Play

Articles described staff permitting children’s adventurous play as a barrier; for example, in Gyllencretuz et al. [40] the extent to which staff allowed children to engage in adventurous play was reported as being judged by staff on a case-by-case basis, and influenced by factors such as child age, development and personality of children, and teachers. Although staff participating in Lester et al.’s [34], study were instructed to “step back” and trust the children to play as part of the intervention, it was apparent that staff were still aware of their responsibilities and in some instances, due to their own personal anxieties, intervened in children’s play.

On the contrary, relaxing rules and permitting children to play was also described as a facilitator. In Farmer et al. [32], school leaders mentioned that relaxing rules meant children had more opportunities to play, and that the play environment was more permissive. Participants in this study also mentioned that fewer rules resulted in teachers stepping back and allowing children more freedom to monitor their own play. Alongside this was an awareness of when staff should interfere with play, as well as a new perspective on safety [32].

## 4. Discussion

This review aimed to provide insights into the perceived barriers and facilitators of adventurous play in schools by bringing together findings from existing qualitative research. We conducted a meta-aggregative synthesis and a narrative synthesis of findings across nine studies. Below, we bring together these findings, reflect on the strengths and weaknesses of the existing literature, and make specific recommendations for policy and practice.

### 4.1. What Are the Perceived Barriers and Faciliatators of Adventurous Play in Schools?

There was considerable consistency between the results of the meta-aggregative synthesis and the narrative synthesis. From a psychological perspective, adults’ perceptions, attitudes, and beliefs about play and about children’s abilities were clearly present in both analyses. Focusing first on adults’ attitudes and beliefs about adventurous play, it was clear across studies that adults often held positive beliefs about the benefits of adventurous play for children, which motivated them to support its provision. However, these attitudes and beliefs existed against a backdrop of uncertainty, which provoked anxiety in supervising children and causing them to intervene in a limiting way. The role of individual differences in perceptions of risk and tolerance of uncertainty was also clear.

Other commonly held perceptions included perceiving children as unable to judge risk and initiate play for themselves, although there were contrasting views about how well children are able to do this independently. There were examples of participants who were able to identify a change in their expectations as they gave children more space to play and recognised that their assumptions were incorrect. This happened when adults consciously decided to step back from children’s play. In doing so, they were able to recognise children’s abilities, thus giving them confidence to step back further [43].

Additionally, consistent between the two analyses was the importance of a whole-school approach to adventurous play, which included parents and school caretakers. Several studies highlighted staff concern about parent reactions, especially if a child could be injured playing adventurously at school. Individual differences in how parents responded to adventurous play opportunities was evident; whilst for some parents this appeared to increase the appeal of the school, in some cases parents chose to remove their child from the school as a result of their approach to play [32]. This contrasting finding highlights the varied perceptions of parents and the challenges schools face in providing opportunities for adventurous play in school. Given this, risk-reframing sessions, which help parents to understand the motivations for this approach to play, are likely to be important. Staff in schools providing this type of risk-reframing session explained that parents attending the sessions gained a mutual understanding that mitigated fears. Achieving this whole-school approach was not straightforward however, with a headteacher in one study describing it as “the hardest challenge” and a school leader in another study reporting that it could be difficult to change the attitudes of caretakers and teachers. Strong leadership and a shared goal appeared to help overcome some of these challenges, as did training and education around adventurous play for all members of the school community. Indeed, Farmer et al.’s [32] study highlights that, without this training, schools may have little knowledge and understanding about how their rules and practices affect children’s play.

In addition to the above, Health and Safety and concerns about legislation were also discussed as barriers. Across several studies, staff mentioned their duty of care as being a barrier to allowing children to take risks when they play. Staff were also concerned about the consequences if things went wrong, including external judgement via outside agencies such as school inspectors or potentially losing their jobs.

### 4.2. Study Reflections and Limitations

The final synthesis consisted of a small number of studies; of the nine studies included in the review, only four met our quality criteria for inclusion in the meta-aggregative synthesis. The primary reason for exclusion of the other papers was that the influence of the researchers on the research (and vice versa) was not clearly acknowledged. This is not to say that these studies were not conducted to a high standard, nor that the results are not informative; rather, what was reported in the available versions of the articles did not allow us to be sure that the quality was high enough for the result to be included in the formal analysis. The overall methodological quality of the articles should be kept in mind when considering the recommendations for policy and practice made below.

We chose to focus the review on qualitative research because this approach provides rich data. This richness allows for a deeper understanding of the issues relevant to the research questions, thus enabling us to make clearer, more specific recommendations for policy and intervention. Although the lack of quantitative data may be considered by some to be a limitation, this depth of understanding is difficult to obtain from quantitative data.

It is noteworthy that several of the studies included within the review focussed specifically on children with disabilities [42,43]. It is plausible that some unique barriers and facilitators of adventurous play may exist for children with disabilities; however, due to the small number of studies identified, a sub-analysis was not possible and we are unable to make recommendations that are specific to children with disabilities. Despite this, many of the findings from the studies that focussed on children with disabilities were represented within core themes that were present across studies. As a result, the recommendations are likely to apply to children with disabilities as well as typically developing children.

A potential limitation is that six of the nine studies included in the review reported data collected as part of an intervention evaluation. The barriers and facilitators identified within the review may therefore more closely reflect experiences of participating in an intervention related to adventurous play. It is plausible that the barriers and facilitators identified outside of an intervention context may differ. Nevertheless, the studies give valuable insights into the mechanisms likely to be involved in supporting and facilitating adventurous play in schools and the barriers that exist for implementing and changing attitudes towards adventurous play in schools. Similarly, the findings are only relevant to school contexts, which aligns with our aims. It is likely that other barriers and facilitators of adventurous play exist within a broader social context. Specific barriers and facilitators may also differ across different countries and cultures; the articles included within this review were primarily from Western countries. Future research would benefit from examining barriers and facilitators of adventurous play across cultures and geographical locations.

### 4.3. Implications for Research

The review and the findings indicate several directions for future research. As aforementioned, six of the nine studies included within the review report data from intervention studies. This suggests that to date, there is relatively little empirical work qualitatively examining the barriers and facilitators of adventurous play in school-aged children that exists to inform interventions. Future work is therefore needed to examine barriers and facilitators of adventurous play outside the context of an intervention. Of the three of the studies included that specifically examined the barriers to and facilitators of adventurous play in schools, outside an intervention context [40,44,45], participants were primarily school professionals, including headteachers and teachers. Given the importance of a whole-school approach within the review findings, it is recommended that research about adventurous play in schools should also include parents, lunchtime supervisors, and the wider school community (e.g., caretakers). Indeed, whilst the necessity for parent support for adventurous play was evident across analyses, the voice of parents pertaining to the barriers and facilitators for adventurous play in schools was absent. Research with the wider school community is critical to gain a wider understanding of the perceived barriers and facilitators of adventurous play in schools.

### 4.4. Recommendations for Policy and Practice

On the basis of our analyses, we make the following recommendations for policy and for practice, specifically in relation to future interventions.

#### 4.4.1. Policy

Regulatory bodies including school inspectors and Health and Safety executives must provide clear guidance regarding risk–benefit analysis and the provision of adventurous play in schools.Funding should be provided to ensure that schools have the resources to ensure children’s play is adequately provided for.

#### 4.4.2. Practice


Interventions must require a whole-school approach, with parents and school staff, including lunchtime supervisors, teachers, and caretakers/cleaners involved and informed.Training and education around adventurous play is vital. Specifically, training must:1.1.Address fears and uncertainty surrounding staff member and school accountability in relation to duty of care and the potential for child injury. This can be gained via clear guidance from regulatory bodies as well as through training.1.2.Include training regarding children’s skills and capabilities to play, including children’s ability to judge risk for themselves.1.3.Include education around how intervening in children’s play and directing through language may limit children’s adventurous play engagement.1.4.Focus on developing positive beliefs about adventurous play, including understanding the benefits of adventurous play.1.5.Include support in how to recognise and evaluate risk and hazards.School staff should be supported to reflect on how their current rules and practices might have a positive and negative impact on children’s play, including what staff do to manage their own uncertainty.Interventions should include some supported practical exercises to carry out which require staff to experiment with stepping back from children’s play and observing what happens. This action of stepping back should facilitate children’s play and provide an opportunity for adults to adjust their perceptions about children’s abilities. Stepping back facilitated children’s ability to play and, therefore, this should be a key message in intervention and training.Interventions must recognise the practical considerations that may arise, such as time and appropriate space, and support schools to overcome these potential barriers.


## 5. Conclusions

The synthesis points to myriad factors that exist in acting as barriers and facilitators in relation to offering adventurous play opportunities and allowing children to engage in adventurous play in schools. Specifically, the findings from this review highlight the importance of adult’s perceptions, attitudes, and beliefs about children and adventurous play, as well as concerns regarding accountability and safety. The findings will inform the design and implementation of future interventions that seek to have a positive impact on children’s global health by increasing their adventurous play at school.

## Figures and Tables

**Figure 1 children-08-00681-f001:**
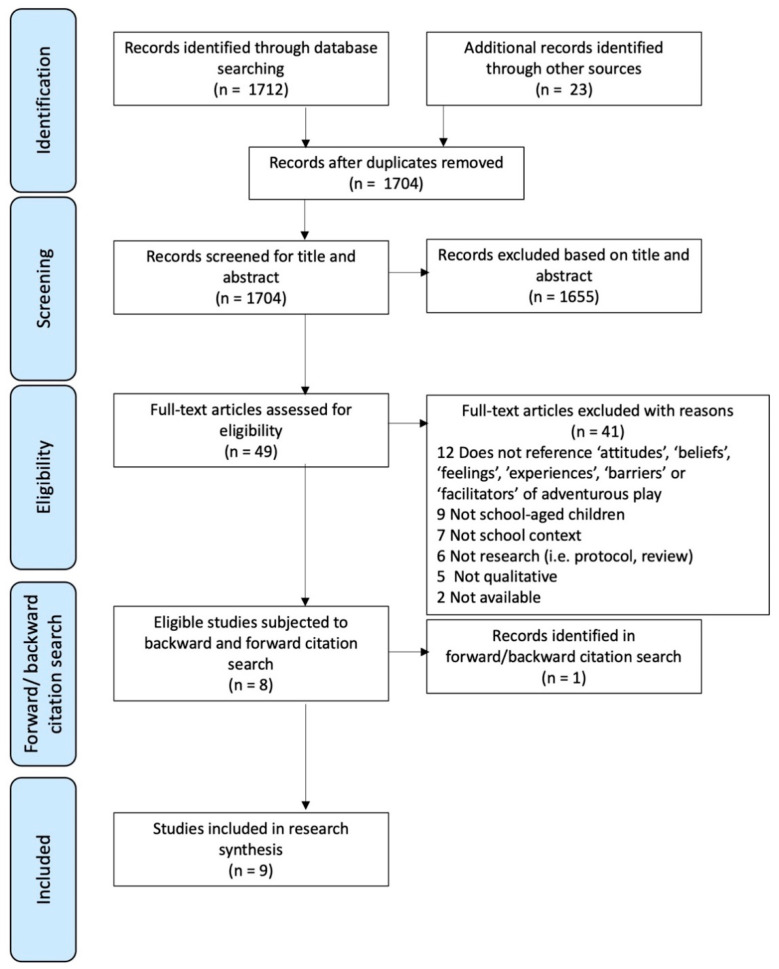
PRISMA diagram, detailing the number of records at each stage of the systematic review process.

**Figure 2 children-08-00681-f002:**
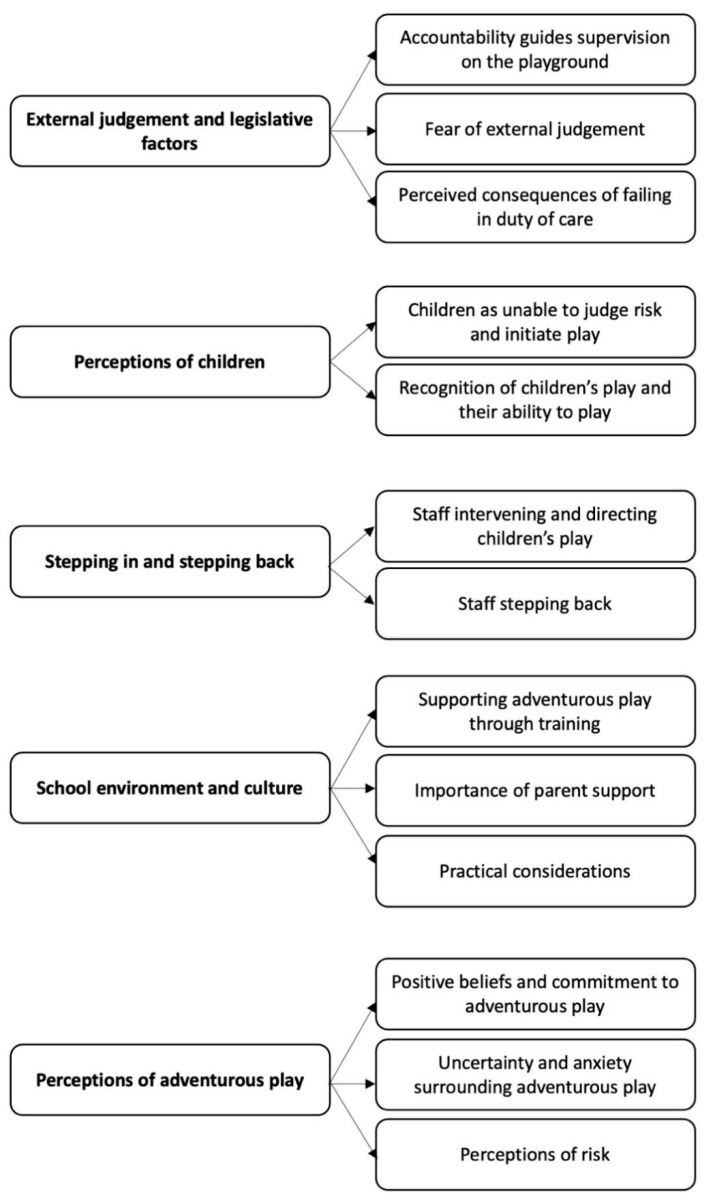
The five synthesised findings and the aggregated categories within.

**Table 1 children-08-00681-t001:** Critical appraisal results for eligible studies using the JBI Qualitative Critical Appraisal Checklist.

	Cevher-Kalburan [39]	Farmer et al. [32]	Gyllencreutz et al. [40]	Lester et al. [34]	Niehues et al. [41]	Spencer et al. [42]	Sterman et al. [43]	Van Rooijen et al. [44]	Wright [45]
1. Is there congruity between the stated philosophical perspective and the research methodology?	Y	Y	Y	Y	U	U	U	U	Y
2. Is there congruity between the research methodology and the research question or objectives?	Y	Y	Y	Y	Y	Y	Y	Y	Y
3. Is there congruity between the research methodology and the methods used to collect data?	Y	Y	Y	Y	Y	Y	Y	U	Y
4. Is there congruity between the research methodology and the representation and analysis of the data?	Y	Y	Y	U	Y	Y	Y	U	Y
5. Is there congruity between the research methodology and interpretation of results?	Y	Y	Y	Y	Y	Y	Y	Y	Y
6. Is there a statement locating the researcher culturally or theoretically?	Y	N	N	Y	N	N	N	N	U
7. Is the influence of the researcher on the research, and vice versa, addressed?	Y	U	N	U	N	Y	Y	N	Y
8. Are participants, and their voices, adequately represented?	Y	Y	Y	Y	Y	Y	Y	N	Y
9. Is the research ethical according to current criteria or, for recent studies, and is there evidence of ethical approval by an appropriate body?	U	Y	Y	U	Y	Y	Y	U	Y
10. Do the conclusions drawn in the research report flow from the analysis, or interpretation, of the data?	Y	Y	Y	Y	Y	Y	Y	Y	Y

Note. Y = Yes, N = No, U = Unclear. Shaded columns = included in meta-aggregative synthesis, unshaded columns = included in the narrative synthesis.

**Table 2 children-08-00681-t002:** Study characteristics of included articles, meta-aggregative synthesis.

Reference	Country	Method	Setting	Participants	Analytical Approach	Phenomenon of Interest
Cevher-Kalburan [39]	Turkey	Questionnaire with open-ended responses, researcher’s reflective notes, participants’ written evaluations, and drawings	Early childhood teacher education programme	26 early childhood pre-service teachers	Content Analysis	Examined the effectiveness of an intervention aimed at changing early childhood pre-service teachers’ understanding of children’s risky play
Spencer et al. [42]	Australia	Field notes and video recordings of structured observations of children’s play; teachers closed and open-ended responses to the Tolerance of Risk in Play Scale (TRiPS)	2 primary schools for children with diverse physical and intellectual special educational needs (high proportion of autistic children)	49 teachers and observations of children’s play	Thematic Analysis	Drew on findings from the Sydney Playground Project to unpack the discomfort experienced by school staff in their responses to uncertain moments in children’s play
Sterman et al. [43]	Australia	Semi-structured interviews	Primary schools (four special schools and one mainstream school with three specialist support classes for children with developmental disabilities)	27 school staff (teaching assistants, teachers, therapists, school leadership) who had participated in the Sydney Playground Project	Thematic Analysis	Examined the utility of the Sydney Playground Project intervention for promoting choice and control among children with disability on the school playground
Wright [45]	UK	Semi-structured interviews and photo-elicitation	3 primary schools	3 headteachers	Thematic Analysis	Examined the attitudes and perceptions of primary school headteachers regarding physical risky play

**Table 3 children-08-00681-t003:** Synthesised finding 1: External judgements and legislative factors.

**Accountability Guides Supervision on the Playground**
▪ “You always have duty of care that takes precedence over everything … I’m accountable to myself in one respect; I’m also accountable to parents. If something happened to a child that would be something I would have to live with” [43].
▪ “our duty of care, responsibility” [42]
**Fear of External Judgement**
▪ “It is the fear factor and often what I hear is, “and what would they say, when they came in?” Who are they? They are afraid of someone coming in and saying, “That is a waste of time”” [45]
▪ “We just thought that people are walking past our school all the time. We don’t want them to think it’s a complete trash heap” [43].
**Perceived Consequences of Failing in Duty of Care**
▪ “I know people have said, “What’s the worst that could happen? He falls and he breaks his arm” But if he fell and broke his arm, we would be in trouble from parents; we would be in trouble from supervisors. “So, we would not let him do that”, you do have the most fun when you’re taking risks, but we still have a duty of care” [43].
▪ “hurting themselves in my care, as I am responsible for someone else’s child”, “them seriously hurting their peers- my responsibilities and having to report to their parents” [42]
▪ “[lunchtime supervisors] fear blame because they have to communicate to other staff and they really are quite stand alone” [45]

**Table 4 children-08-00681-t004:** Synthesised finding 2: Perceptions of children.

**Children as Unable to Judge Risk and Initiate Play**
▪ “Most of our kids are very good at unstructured activities, [but] it’s not play. It tends to be repetitive movements or speaking or doing a routine over and over in the playground. If there’s not someone to make something interesting, then that’s what a lot of our guys will do” [43]
▪ “Their inability to imagine what dangers are present or how they may affect them is a great fear. They have limited ability to solve or generalise dangers” [42]
▪ “There is a reasonably high level of acceptable risk-taking, but it is definitely balanced with a real understand that our staff have of duty of care to the students, and they mustn’t let them do something where they’re going to get hurt. Particularly [these] children who are more vulnerable and may not understand the consequences of unsafe actions that they undertake. It is a real mindset, and it does limit risk taking” [43]
**Recognition of Children’s Play and Their Ability to Play**
▪ “It was definitely interesting to see some of the kids who usually don’t engage with our play equipment engaging with something” [43]
▪ “I realized that I substituted risk and hazard with each other previously” ... “But now I know that risk can be assessed by children if we give them this opportunity” [39]
▪ “we saw a lot of really cool stuff happen that we didn’t realise those kids would or could do” [43]

**Table 5 children-08-00681-t005:** Synthesised finding 3: Stepping in and stepping back.

**Staff Intervening and Directing Children’s Play**
▪ (Video extract) “A child was climbing on the play structure when a member of school staff joined her. The teacher asked a colleague about the rules with regards to how high the child should climb. The colleague responded by suggesting that the child appeared to be steady, but the child should be watched to ensure her safety. Soon after, the teacher lifted the girl off the play structure and redirected her to play elsewhere” [42]
▪ “I hear a lot of be careful, I think it trips off our tongue, I think what we need to think about is not directing children to be careful but, what do we need to think about?” [45]
**Staff Stepping Back**
▪ “there were a lot of things that I seriously thought the kids would not be interested in, but they were interested in. I think it made us push ourselves a little bit in letting go cause we’re holding on and keeping them safe. It just made you step back and say “Okay, they can do it. Just let them do it”. You saw that they do it on their own if you give them the opportunity and not step in and say ‘Oh let me help you” giving them more independence from us” [42]
▪ “I encourage them to learn and engage in activities independently, always let them have a go first” [43].
▪ “With the long noodles, they began using [them] as swords. I’d wait over there and have my heart palpitating going, “Oh my gosh.” But until I actually took that step back I [didn’t] realise “oh, that’s how they play”. As long as they’re not physically hurting each other they’re okay. It’s definitely changed the way that I supervise those kids” [43].

**Table 6 children-08-00681-t006:** Synthesised finding 4: School environment and culture.

**Supporting Adventurous Play through Training**
▪ “Just to empower them [lunchtime supervisors] to see things more positively and managing groups of people without having them [the children] standing as if they are on parade” [45] ▪ “We are taught, and our new teachers are taught about filling in and looking at risk in terms of what it really means and what you need to look out for that could become barriers. The form and process are an enabler it is just need to be aware of to make it a success not what could go wrong and lead to danger” [45]
▪ “He [Local Authority Children and Schools Health, Safety and Wellbeing Manager] is very much not a barrier, he is very must promoting risky play and activities, an enabler” [45]
**Importance of Parent Support**
▪ “It was so helpful when the parents said, oh we understand kids hurt themselves all the time, it’s not a huge concern of ours” [43]
▪ “It would have been a lot better if we’d had parents there. I think that was a key miss for us” [43]
▪ “We need to have the trust and understanding of parents and families” [45]
**Practical Considerations**
▪ “It is using timetable time when you have every other aspect of the curriculum to cover as well; it is finding time in the timetable to do it” [45]
▪ “for example, these (Image 5) get really slippery then it is wet, you probably would not let children get onto a high one of those when it is really wet because they will slip” [45]

**Table 7 children-08-00681-t007:** Synthesised finding 5: Perceptions of adventurous play.

**Positive Beliefs and Commitment to Adventurous Play**
▪ “before this course, I viewed risk differently than now … But now I am aware that they need to do such things to grow up healthy and develop many skills” [39] ▪ “You have got to believe in risky, active play, you have got to have a total commitment as to why you want to do it, what you believe are the benefits for children. If you are not committed to it, then I do not really see it working” [45].
**Uncertainty and Anxiety Surrounding Adventurous Play**
▪ “they are not quite sure if the play is moving into an unacceptable level of behaviour or dangerous play or whether it is just high spirits” [45]. ▪ “we’re trying not to model or get too much with the students if they were interest acting with, because they weren’t really sure what they could do or what level of modelling they could provide” [43]. ▪ “there is your natural nervous adult, who would rather not take the risk themselves and therefore would not have these opportunities happening at all” [45].
**Perceptions of Risk**
▪ “Before this course my risk perception was superficial. I realized that I overused “risky” term regarding children’s play… Now I am aware of what exactly risk and hazard are” [39] ▪ “Life is a risk and this is a skill they need to come across and learn to deal with in their own ways. If they do not have the opportunity, how are they ever going to deal with the adrenalin?” [45]. ▪ “Because [the fixed equipment] is an accepted object, people don’t really think about that as a risk” [43].

**Table 8 children-08-00681-t008:** Study characteristics of included articles, narrative synthesis.

Reference	Country	Method	Setting	Participants	Analytical Approach	Phenomenon of Interest
Farmer et al. [32]	New Zealand	Interviews and field notes	8 primary schools	10 interviews with school leaders	Thematic Analysis	Examined the acceptability of an intervention designed to increase risk and challenge in the school playground
Gyllencreutz et al. [40]	Sweden	Observations and focus groups (4 with teachers and 6 with children)	2 urban primary schools	28 teachers and 48 children (24 pre-school (6–7 years) and 24 fifth grade (11–12 years) children) participated in focus groups	Content Analysis	Investigated risky outdoor play within the school playground and teachers’ perceptions of risk and safety in relation to learning and development
Lester et al. [34]	UK	Phase 1: document review, telephone interviews. Phase 2: case study (observations, focus groups and interviews)	Phase 1: 29 schools; primary, infant, and junior schools (10 interviewed). Phase 2: 3 schools	Telephone interviews (headteachers). Case study (interviews and focus groups with headteachers, teaching staff, and lunchtime supervisors). Observations of children’s use of the outdoor areas	Thematic Analysis	Examined the effectiveness of OPAL in improving play opportunities for children in schools and how schools benefit from participating in OPAL
Niehues et al. [41]	Australia	Risk-reframing groups	9 primary schools	150 parents and school staff and community agency volunteers	Social analysis	Examined the effectiveness of a child-centred risk-reframing intervention in altering adults’ perceptions of risk
Van Rooijen et al. [44]	The Netherlands	Questionnaire with open-ended responses	Childcare environments	59 professionals working in childcare environments (48 in childcare organisations, 6 in primary education, and 5 in other, e.g., SEN environments, retired)	Content analysis	Examined whether challenges identified within Van Rooijen and Newstead’s (2016) models impact children’s risk-taking play in Dutch childcare contexts

## Data Availability

Not applicable.

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
