# Peer review of "Perceived Barriers and Facilitators of Adventurous Play in Schools: A Qualitative Systematic Review"

_children, 2021, doi:10.3390/children8080681_

Round 1
Reviewer 1 Report
This is a well written article. Review protocol clearly and rigorously described and reported. Identification of the key themes arising from both the meta-aggregative synthesis and narrative synthesis are well organised and discussed. Clear and useful recommendations for policy and practice identified. The article has potential to inform efforts aimed at greater adoption of adventurous play in schools.
Author Response
We would like to thank the reviewer for their positive review of our manuscript.
Reviewer 2 Report
The article presents an interesting contribution to the understanding of adventurous and risky play. I congratulate the authors for this work.
In the theoretical framework the authors should have dealt in more depth with some aspects of interest and necessary to favour the link between the theoretical framework and the discussion of the results.
Health.
The concept of health refers to different dimensions: organic (e.g. sedentary lifestyle, overweight, improvement of physical capacities: coordination, laterality...), affective (emotional), relational (social) and cognitive (decision making).
Defending the use of play to promote health should develop in more detail the importance of play for the levels indicated. The authors only talk about physical and mental health. However, during the discourse they mention some emotions or feelings such as happiness, joy, anxiety.
Moreover, it could be linked to the fact that people who play are in a state of flow, where their emotional and sensitive rather than rational intervention leads them not to value the risk of their actions.
In line 760 they talk about emotional wellbeing and physical health, which confirms the need to consider these aspects in the theoretical framework.
On other occasions they talk about behaviour, health and development. These three aspects should be developed in more detail.
Play, rules
Line 57. They talk about unstructured play. There should be some mention of the type of activities referred to as play with or without rules and their link to the expression of unstructured play activity.
Family and adults (staff, teachers) in play.
The role of the family and educators in children's play should be discussed in more detail in the theoretical framework. Also the situation in relation to the role of the family in education in and out of school.
The beliefs of adults in relation to play.
Conclusions
Try to use a global concept of health , not just physical health (line 761).
Author Response
The article presents an interesting contribution to the understanding of adventurous and risky play. I congratulate the authors for this work.
1. In the theoretical framework the authors should have dealt in more depth with some aspects of interest and necessary to favour the link between the theoretical framework and the discussion of the results.
-
- The concept of health refers to different dimensions: organic (e.g. sedentary lifestyle, overweight, improvement of physical capacities: coordination, laterality...), affective (emotional), relational (social) and cognitive (decision making). Defending the use of play to promote health should develop in more detail the importance of play for the levels indicated. The authors only talk about physical and mental health. However, during the discourse they mention some emotions or feelings such as happiness, joy, anxiety.
- Moreover, it could be linked to the fact that people who play are in a state of flow, where their emotional and sensitive rather than rational intervention leads them not to value the risk of their actions.
- In line 760 they talk about emotional wellbeing and physical health, which confirms the need to consider these aspects in the theoretical framework.
- On other occasions they talk about behaviour, health and development. These three aspects should be developed in more detail.
We agree with the reviewer that health is a broad concept and that our initial introduction was too narrow in its focus (physical and mental health only). We have now added a general paragraph to the introduction, outlining how play may be beneficial for children’s health more broadly, encompassing the different dimensions of health the reviewer noted (physical activity, physical capability, affective skills, social skills and cognitive functioning).
When we discuss adventurous play we still focus on physical and mental health because there is lack of theoretical work linking adventurous play to other health outcomes. We have now acknowledged this in the introduction and have noted that whilst theoretical work is scarce, there is a growing body of empirical work linking adventurous play to other health outcomes in children.
We agree that our original description of the findings of the systematic review on risky play and health outcomes was perhaps too vague and have now included some specific findings from the review, using their wording, to support the importance of risky/adventurous play for health outcomes, broadly defined.
2. Play, rules: Line 57. They talk about unstructured play. There should be some mention of the type of activities referred to as play with or without rules and their link to the expression of unstructured play activity.
We have added a definition of what is meant by unstructured play in this context; play that is intrinsically motivated, spontaneous, without a set purpose and in the absence of external rewards.
3. Family and adults (staff, teachers) in play: The role of the family and educators in children's play should be discussed in more detail in the theoretical framework. Also the situation in relation to the role of the family in education in and out of school. The beliefs of adults in relation to play.
We agree that there was little included in our theoretical framework outlining the importance of family and educators in children’s play. We have edited the introduction to add in more detail here, specifically we draw upon Dodd and Lester’s (2021) theoretical model, describing the importance of the social environment in facilitating adventurous play. We also draw upon some research on attitudes towards play and associations with how much time children spend playing adventurously.
4. Conclusions: Try to use a global concept of health, not just physical health (line 761).
We have changed the conclusion to use the term global health instead of just physical health.